# Social and natural risk factor correlation in China's fresh agricultural product supply

**Xiaoyu Bian**[1], **Guanxin Yao**[2]\*, **Guohong Shi**[1]

**1** School of Management, Jiangsu University, Zhenjiang, Jiangsu, China, **2** School of Business, Yangzhou University, Yangzhou, Jiangsu, China

\* 827172632@qq.com

## Abstract

The perishable nature of fresh agricultural products and their vulnerability to environmental impacts make fresh agricultural product supplies susceptible to more complex social risks and unpredictable natural risks. This study identifies 13 social and natural risk factors that could adversely affect the fresh agricultural product supply and uses ISM and MICMAC to develop a hierarchical structure of the risks to analyze the correlation of these risk factors. The results showed that these risk factors have a strong positive correlation, a reasonable risk-sharing mechanism should be established for the fresh agricultural product supply and the improvement in the supervision error correction system should be strengthened.

## Introduction

In the last two decades, the trends of changing consumer demand, economic globalization, increasing government regulations, severe environmental pollution, and significant fresh agriculture product safety scares in China have led to a profound increase in concerns over fresh agricultural product supply risk. Supply risk is increasingly referred to as an important concept related to the fresh agricultural product supply system. However, the concept of supply risk is not well defined or understood. The fresh agricultural product supply risk debate has become a central issue in both academic and industry circles [1–3]. However, the inconsistent usage and definitions of supply risk are affecting this central issue. Thus, clarifying what the supply risk of fresh agricultural products means is an urgent problem.

The supply process of fresh agricultural products includes the purchase of agricultural materials, production of fresh agricultural products, transportation and storage of fresh agricultural products, and sales of fresh agricultural products in China, and this supply process involves multiple participants, long production cycles, delayed production planning, the perishable nature of fresh agricultural products and the characteristics of less flexible supply and demand. This process is vulnerable to political, economic, and legal issues as well as the natural environment and other factors. As a result, the supply of fresh agricultural products faces extensive risk in China. In short, fresh agricultural products are a combination of natural products and socioeconomic production processes. Therefore, unlike industrial companies, the fresh agricultural product supply is simultaneously subject to multiple natural and social risks, leaving the fresh agricultural product supply in a vulnerable position, and fresh agricultural

**Funding:** The research was supported by National Natural Science Foundation of China (No. 71473213 and 71773104) to GY. The funder had

no role in study design, data collection and analysis, decision to publish, or preparation of the manuscript.

**Competing interests:** The authors have declared that no competing interests exist.

product suppliers need to pay attention to the natural environment of production and the quality and freshness of fresh agricultural products. If these social and natural risks cannot be properly identified and controlled, then the poor situation of China's fresh agricultural product supply will be difficult to improve.

## Literature review

Research seeking to guide the development of agricultural product supply risk management is varied, and most scholars use models to illustrate a specific agricultural product supply risk and use case studies to highlight their analyzes. Kingwell *et al.* [4] built a model (named MUDAS) to demonstrate the effect of climate risk on the agricultural product supply. Nugrahani [5] presented some mathematical models to model financial risks related to the agricultural product supply, which shows that the agricultural business is quite a risky business for suppliers. Severini *et al.* [6] modeled agricultural risk management policies to reduce farmers' income risk using Italian agriculture as a case study. Nitu *et al.* [7] claimed that agricultural risks are related to water regime disturbance. Thus, they proposed an optical spectral image recognition algorithm, and examples of simulation experiments with Global Information-Modelling System (GIMS)-based tools are given. Musshoff *et al.* [8] applied real yield and weather data from Northeast Germany to quantify the risk-reducing effect that can be achieved in wheat production by using precipitation options. To do so, a stochastic simulation was used. The hedging effectiveness is controlled by the contract design (index, strike level, tick size). Turvey and Driver [9] developed a Farm Sector Capital Asset Pricing Model (FSCAPM) to examine systematic agricultural risks, and the results revealed that for many agricultural commodities and crop mixes, the amount of systematic risk is high.

Again, some scholars focused on concluding survey data, using financial analyzes and econometrics as research methods. Liu and Miller [10] combined econometric and financial analyzes using the NAHMS 2000 Swine Survey data and found sufficient evidence that the use of antibiotics for growth promotion (AGP) can reduce risks and increase profits in the U.S. swine industry. Li [11] illustrated the impact of pesticides, fertilizers, plastic films and the aquaculture industry on agricultural pollution risk and believed that agricultural pollution was caused by the combined failures of government regulation and the market. Using an advanced very high-resolution radiometer (AVHRR), Qian [12] obtained vegetation status index (VCI) data on agricultural areas in China from 1982 to 2010 and analyzed the temporal and spatial changes of each drought and the relationship between drought and climate risk factors. Sarwar [13] conducted detailed interviews and questionnaires to study how Pakistani farmers perceive and manage risk and proposed that the research findings can help to develop an integrated risk management strategy framework. Considering the Intergovernmental Panel on Climate Change's (IPCC's) assessment of major risks for African agriculture, despite great uncertainty, and the published literature, Mueller [14] made several robust conclusions for policy-makers and research agendas: agriculture everywhere in Africa faces some risk of being negatively affected by climate change; existing cropping systems and infrastructure will have to change to meet future demand. Meuwissen [15] studied risk management strategies such as insurance (in which risks are shared with others) to find out whether such strategies provide opportunities for farmers to deal with new risks confronting agriculture, and they concluded that risk-sharing strategies do provide such opportunities on both theoretical and empirical grounds.

Several studies have begun to document the correlation between social risk factors and the agricultural product supply, and these largely resonate with our study. Wang [16] used the fuzzy comprehensive evaluation method to evaluate the risk of an agro-product closed supply chain, and the results showed that the key first-level indices influencing the risks are system

risk, information risk, quality, and safety risk, and the key second-level indices are the coordinating and controlling ability of core enterprises. Laura [17] described the main types of risk in agriculture, their features and their relevance concerning scientific theories; introduced the most popular risk evaluation methods and their potential use in agriculture; and finally, presented a logical framework of integrated risk evaluation.

The aforementioned documents mainly show that the main obstacle to the development of fresh agricultural product supply is difficulties in avoiding and sharing social and natural risks because these risks are more complex and unpredictable than in other industries. We draw on some available research findings from these papers and obtain some inspiration from them. However, the object of this study is the fresh agricultural product supply. Unlike other agricultural product suppliers, fresh agricultural product suppliers pay more attention to agricultural product timeliness and quality, so more social and natural risks need to be avoided, for example by using refrigeration technology, fresh-keeping treatment, transportation, and storage. This is a difference from the above research that cannot be ignored. Second, the above-mentioned studies are not comprehensive and meticulous in identifying and categorizing risks faced by the fresh agricultural product supply, nor do they distinguish the levels of risks, and these studies involve only production, management and sales and do not consider agricultural material purchases. Therefore, generalized risk countermeasures cannot be applied to support the fresh agricultural product supply, and we need to do much more work to correctly identify natural and social risk correlations based on the previous literature.

In the following sections, we first identify the risk factors of the fresh agricultural product supply through expert opinion and a literature review, allowing us to finally summarize 13 core risk factors. Then, we establish a model to transform the chaotic risk factor relationship into an intuitive relationship structure model. Second, we establish an interpretive structural model (ISM) that transforms chaotic risk factor relationships into intuitive relationship structure models and then use MICMAC to classify the risk factors according to their driving forces and dependencies to obtain detailed analysis results of the relevance and importance of the risk factors. Finally, we summarize some general policy insights obtained from the analysis results and propose future research prospects.

## Problem description

According to the National Bureau of Statistics of China (NBSC), the fresh agricultural supply plays an important role in agricultural economic development in China and is also the main source of farmers' income. In past decades, the Chinese economy has developed well, and residents' income has increased rapidly. In 2013, per capita, consumption expenditures reached 1,8311 yuan in China, and by 2019, they reached 3,0733 yuan, showing that the purchasing power of residents is increasing exponentially, and this growth leads to great pressure on the supply of fresh agricultural products because residents expect more varieties and high-quality fresh agricultural products.

Moreover, China's fresh agricultural product supply is poor compared to that of other developed countries, but it has many opportunities for improvement. The fresh agricultural product supply of China has always suffered from overburdened variety development, product preservation, environmental pollution, and a lack of infrastructure and production technology. The main reason for the failure of China's fresh agricultural product supply is that it bears more economic and natural risks than other industrial product supplies, resulting in a decline in the initiative of fresh agricultural product suppliers and an unwillingness to invest more in the supply of fresh agricultural products. Therefore, the problem addressed in this article is the need to identify the correlations of the risk factors of the supply of fresh agricultural products in China.

## Social and natural risk factors in the fresh agricultural product supply

### Methodology of identifying the social and natural risk factors

In the present study, we obtained research data mainly by collecting and analyzing textual data obtained from Elsevier, China National Knowledge Infrastructure (CNKI), Emerald, Scopus, etc. Textual data were found in the literature using the keywords "fresh agricultural products" and "risk" or "invalid" from 2010 to 2019. However, the number of research papers directly associated with the concept of fresh agricultural products is small. To conduct a comprehensive review of the literature, all the research papers related to the concepts of fresh food, fresh products, vegetables, meat, etc. were added to the scope of the literature search.

In the end, more than 200 research papers were searched. Interestingly, these results also include some irrelevant research, namely, research on fresh-water, cold chain logistics, sustainability, agricultural economics, and industrial management, which was excluded from the review process.

### The identification of the social and natural risks of the fresh agricultural product supply

After these related studies were searched and classified, we abstracted possible initial risk factors of the fresh agricultural product supply as comprehensively as possible, and the Delphi technique was used to summarize these initial factors into core risk factors. A description of the selection process of core risk factors is provided in the following:

(1) Relevant data were collected for textual analysis, information was extracted according to keyword risk and failure, and the initial risk factors were determined.

(2) Members of the expert group were identified.

(3) Individual opinions on the initial risk factors were gathered, and the opinions from the expert group were analyzed.

(4) Information on new comprehensive opinions was compiled and sent to each member of the expert group for review.

(5) Each expert was asked to revise their own opinions based on the responses from the expert group. If the individual's response was significantly different from that of the group norm, the member needed to provide a rationale for his differing viewpoint.

(6) Analyzing the new individual opinions and returning to the members of the responses.

(7) This analysis process was repeated many times to gradually obtain a more consistent result.

A total of 15 relevant professors and experts from Yangzhou University, Nanjing University and Jiangsu University, all of whom had conducted research on agricultural risk, and other relevant personnel who had engaged in fresh agricultural production and supply were invited to participate in the process using the Delphi technique. Finally, 13 core categories were extracted, reorganized, integrated, and further classified from the initial risk factors, based on the more consistent results of the Delphi technique. Table 1 shows the 13 core risk factors.

### Research methods

Interpretive structural modeling (ISM) was first proposed by J. N. Warfield [18] to analyze complex socioeconomic systems. ISM is an interactive management tool, namely, an interactive learning process and systematic application of a graphical method that efficiently constructs a directed graph and contextual relationship among a set of elements connected through a variety of circumstances. ISM represents the information either by a direct graph or

**Table 1. Risk factors of the fresh agricultural product supply.**

| R1 | Completion risk | Due to improper production behavior, unreasonable storage and transportation, and insufficient resource endowment, the quantity and quality of fresh agricultural products failed to meet expected standards as scheduled. |
|---|---|---|
| R2 | Information distortion risk | Due to the lengthy process of information transmission, more interference information, information loss, and other reasons, key problems regarding information distortion cause a dilemma for the fresh agricultural product supply. |
| R3 | Financial risk | Due to rising costs, difficulties in charging fees, changes in the prices charged, breaks in the capital chain, etc., the supply of fresh agricultural products cannot reach the expected level. |
| R4 | Technology and management risk | Due to unreasonable planning, immature agricultural technology, and poor corporate management, the supply of fresh agricultural products cannot reach the expected level. |
| R5 | Material equipment risk | The agricultural materials, packaging, transportation vehicles, storage facilities, etc. required for the operation are not up to the standard in terms of quantity or quality, thereby distressing the business. |
| R6 | Regulation risk | Due to the promulgation or amendment of regulations, the fresh agricultural product supply is in a difficult position in terms of operation, affecting the level of fresh agricultural product supply. |
| R7 | Artificial risk | Malicious destruction, operational errors, low-quality employees and other factors affect the supply of fresh agricultural products. |
| R8 | Market risk | Due to factors such as the macroeconomy, the social environment, and adjustments in-laws and regulations, market demand changes. |
| R9 | Pollution risk | Due to factory pollution or the excessive use of chemical fertilizers, the production environment made up of soil, water, air and other factors is polluted, affecting the production of fresh agricultural products. |
| R10 | Biological disease risk | Due to biological diseases, the output of fresh agricultural products is reduced or even cut off. |
| R11 | Natural climate risk | Due to drought, floods, hurricanes, hail and other climatic disasters, the output of fresh agricultural products is reduced or even cut off. |
| R12 | Production risk | Problems occur in the production of fresh agricultural products due to unreasonable production behavior, production technology, etc. |
| R13 | Logistics risk | Problems arise in the transportation and storage of fresh agricultural products due to unreasonable behavior, immature technology and other reasons. |

by a matrix. The ISM model portrays the hierarchy of the variables and the structure of complex issues being studied. Raj *et al.* [19] presented the following characteristics of ISM:

(1) This methodology is interpretive, as the final model reveals whether and how associations exist between different elements.

(2) ISM is a modeling technique because it ultimately depicts specific relationships and the overall structure in a digraph model.

(3) ISM can impose order and direction on complex relationships that exist among various elements.

(4) The ISM approach is mainly a group learning process but can also be used individually [20].

(5) The main limitation of ISM is that the relationships among the variables depend entirely on the knowledge and industry experience of the judges. Therefore, the bias of judges may affect the final results.

The following steps are involved in ISM techniques [21–22].

Step 1. Identification of the variables relevant to the issues or problems with the help of expert opinion and a literature review.

Step 2. From the variables identified in step 1, a contextual relationship is identified among the variables to determine which pairwise variables should be checked.

Step 3. From the textual relationship among variables identified in step 2, a structural self-interaction matrix (SSIM) is developed for the variables of the system under consideration and shows the relationship among pairwise variables.

Step 4. A reachability matrix (RM) is constructed from the structural self-interaction matrix created in step 3, and the matrix is checked for the transitivity of the contextual relation, which is a basic assumption made in ISM.

Step 5. The reachability matrix obtained in step 4 is partitioned into different levels.

Step 6. Based on the relationships given in the abovementioned reachability matrix, a directed graph is drawn, and the transitive links are removed.

Step 7. The ISM digraph is developed. The final ISM model is reviewed to check for conceptual inconsistency, and necessary modifications are made.

Step 8. MICMAC analysis is applied to the model.

## Application of ISM-MICMAC

### Building an interpretive structure model

**Structural self-interaction matrix (SSIM).** SSIM is an ISM methodology used to identify the contextual relationships among the 13 core risk factors identified based on the literature review and expert opinion. The contextual relationship indicates that one variable leads to another variable (i and j), namely, a relation exists between any two variables (i and j) and can be used to identify the associated direction of the relation. In this paper, four symbols are used to denote the direction of the relationships between the variables (i and j):

(1) V: risk factor i will alleviate risk factor j

(2) A: risk factor j will alleviate risk factor i

(3) X: risk factor i and risk factor j will alleviate each other

(4) O: risk factor i and risk factor j are unrelated

Table 2 shows the results of the SSIM.

**Reachability matrix (RM).** First, the adjacency matrix is established and the relationships between the risk factors that affect the fresh agricultural product supply are assigned either 1 or 0 depending on if a relationship exists or not, respectively, to quantify the qualitative relationships [23]. A matrix of these elements is constructed. Taking the definition of the structural model of risk interpretation as an example, the adjacency matrix is named A, and the

**Table 2. Structural self-interaction matrix (SSIM).**

|  | R13 | R12 | R11 | R10 | R9 | R8 | R7 | R6 | R5 | R4 | R3 | R2 | R1 |
|---|---|---|---|---|---|---|---|---|---|---|---|---|---|
| R1 | A | A | A | A | A | O | A | A | A | A | A | A | - |
| R2 | V | V | O | O | V | V | A | O | O | O | O | - |  |
| R3 | V | V | O | V | V | O | O | O | V | V | - |  |  |
| R4 | V | V | O | V | V | O | A | O | O | - |  |  |  |
| R5 | V | V | O | V | V | O | A | O | - |  |  |  |  |
| R6 | O | O | O | O | O | V | O | - |  |  |  |  |  |
| R7 | V | V | O | V | V | V | - |  |  |  |  |  |  |
| R8 | O | O | O | O | O | - |  |  |  |  |  |  |  |
| R9 | O | O | O | O | - |  |  |  |  |  |  |  |  |
| R10 | O | O | O | - |  |  |  |  |  |  |  |  |  |
| R11 | O | O | - |  |  |  |  |  |  |  |  |  |  |
| R12 | O | - |  |  |  |  |  |  |  |  |  |  |  |
| R13 | - |  |  |  |  |  |  |  |  |  |  |  |  |

mathematical definition is as follows:

$$A = \left(a_{ij}\right)_{n \times n}$$

$$a_{ij} = \begin{cases} 1, S_i R S_j \\ 0, S_i \bar{R} S_j \end{cases}$$

which can be proved by the Boolean matrix algorithmml:

$$(A + I)^2 = I + A + A^2$$

Similarly, it can be proved that

$$(A + I)^k = I + A + A^2 + A^3 + \ldots + A^k$$

If system A meets the conditions

$$(A + I)^{k-1} \neq (A + I)^k = (A + I)^{k+1} = M$$

then M is the reachability matrix of the system. The reachability matrix M is calculated by a MATLAB program that has been written. Table 3 shows the results of the calculation.

From the reachability matrix M, we can obtain the antecedent and reachability set of each risk factor [24] and the intersections of the antecedent and reachability sets of all risk factors. The reachability set includes the risk factor itself and other risk factors that it may intensify. The antecedent set consists of the risk factor itself and the other risk factors that might intensify the focal factor.

The risk factor for which the reachability and the intersection sets are the same is identified as the top-level variable in the ISM hierarchy. After the top-level elements are identified, these elements are extracted from the other remaining risk factors [25], and then, the same method is applied again to determine the elements of the second level. From Tables 4–7, it can be seen that the identification process of these risk factors is completed in four iterations.

It can be seen from Table 4 above that the first-level element set of the ISM includes R1 and R8. These elements are removed from the reachability matrix to find the second-level reachability and antecedent sets.

**Table 3. Reachability matrix.**

|     | R1 | R2 | R3 | R4 | R5 | R6 | R7 | R8 | R9 | R10 | R11 | R12 | R13 |
|-----|----|----|----|----|----|----|----|----|----|-----|-----|-----|-----|
| R1  | 1  | 0  | 0  | 0  | 0  | 0  | 0  | 0  | 0  | 0   | 0   | 0   | 0   |
| R2  | 1  | 1  | 0  | 0  | 0  | 0  | 0  | 1  | 1  | 0   | 0   | 1   | 1   |
| R3  | 1  | 0  | 1  | 1  | 1  | 0  | 0  | 0  | 1  | 1   | 0   | 1   | 1   |
| R4  | 1  | 0  | 0  | 1  | 0  | 0  | 0  | 0  | 1  | 1   | 0   | 1   | 1   |
| R5  | 1  | 0  | 0  | 0  | 1  | 0  | 0  | 0  | 1  | 1   | 0   | 1   | 1   |
| R6  | 1  | 0  | 0  | 0  | 0  | 1  | 0  | 1  | 0  | 0   | 0   | 0   | 0   |
| R7  | 1  | 1  | 0  | 1  | 1  | 0  | 1  | 1  | 1  | 1   | 0   | 1   | 1   |
| R8  | 0  | 0  | 0  | 0  | 0  | 0  | 0  | 1  | 0  | 0   | 0   | 0   | 0   |
| R9  | 1  | 0  | 0  | 0  | 0  | 0  | 0  | 0  | 1  | 0   | 0   | 0   | 0   |
| R10 | 1  | 0  | 0  | 0  | 0  | 0  | 0  | 0  | 0  | 1   | 0   | 0   | 0   |
| R11 | 1  | 0  | 0  | 0  | 0  | 0  | 0  | 0  | 0  | 0   | 1   | 0   | 0   |
| R12 | 1  | 0  | 0  | 0  | 0  | 0  | 0  | 0  | 0  | 0   | 0   | 1   | 0   |
| R13 | 1  | 0  | 0  | 0  | 0  | 0  | 0  | 0  | 0  | 0   | 0   | 0   | 1   |

**Table 4. Division of the first-level factors.**

| si | R(si) | Q(si) | R(si) ∩ Q(si) |
|---|---|---|---|
| 1 | 1 | 1, 2, 3, 4, 5, 6, 7, 9, 10, 11, 12, 13 | 1 |
| 2 | 1, 2, 8, 9, 12, 13 | 2, 7 | 2 |
| 3 | 1, 3, 4, 5, 9, 10, 12, 13 | 3 | 3 |
| 4 | 1, 4, 9, 10,12, 13 | 3, 4, 7 | 4 |
| 5 | 1, 5, 9, 10, 12, 13 | 3, 5, 7 | 5 |
| 6 | 1, 6, 8 | 6 | 6 |
| 7 | 1, 2, 4, 5, 7, 8, 9, 10, 12, 13 | 7 | 7 |
| 8 | 8 | 2, 6, 7, 8 | 8 |
| 9 | 1, 9 | 2, 3, 4, 5, 7, 9 | 9 |
| 10 | 1, 10 | 3, 4, 5, 7, 10 | 10 |
| 11 | 1, 11 | 11 | 11 |
| 12 | 1, 12 | 2, 3, 4, 5, 7, 12 | 12 |
| 13 | 1, 13 | 2, 3, 4, 5, 7, 13 | 13 |

**Table 5. Division of the second-level factors.**

| si | R(si) | Q(si) | R(si) ∩ Q(si) |
|---|---|---|---|
| 2 | 2, 9, 12, 13 | 2, 7 | 2 |
| 3 | 3, 4, 5, 9, 10, 12, 13 | 3 | 3 |
| 4 | 4, 9, 10, 12, 13 | 3, 4, 7 | 4 |
| 5 | 5, 9, 10, 12, 13 | 3, 5, 7 | 5 |
| 6 | 6 | 6 | 6 |
| 7 | 2, 4, 5, 6, 7, 9, 10, 12, 13 | 7 | 7 |
| 9 | 9 | 2, 3, 4, 5, 7, 9 | 9 |
| 10 | 10 | 3, 4, 5, 7, 10 | 10 |
| 11 | 11 | 11 | 11 |
| 12 | 12 | 2, 3, 4, 5, 7, 12 | 12 |
| 13 | 13 | 2, 3, 4, 5, 7, 13 | 13 |

**Table 6. Division of the third-level factors.**

| si | R(si) | Q(si) | R(si) ∩ Q(si) |
|---|---|---|---|
| 2 | 2 | 2, 7 | 2 |
| 3 | 3, 4, 5 | 3 | 3 |
| 4 | 4 | 3, 4, 7 | 4 |
| 5 | 5 | 3, 5, 7 | 5 |
| 7 | 2, 4, 5, 7 | 7 | 7 |

**Table 7. Division of the fourth-level factors.**

| si | R(si) | Q(si) | R(si) ∩ Q(si) |
|---|---|---|---|
| 3 | 3 | 3 | 3 |
| 7 | 7 | 7 | 7 |

It can be seen from Table 5 above that the second-level element set of the ISM includes R6, R9, R10, R11, and R8. These elements are removed from the reachability matrix to find the third-level reachability and antecedent sets.

It can be seen from Table 6 above that the third-level element set of the ISM includes R2, R4, and R5. These elements are removed from the reachability matrix to find the fourth-level reachability and antecedent sets.

It can be seen from Table 7 above that the fourth-level element set of the ISM includes R3 and R7.

**ISM digraph.** After four iterations, all risk factors are placed in their exact position and identification level. According to the above results, the reachability matrix is rearranged after removing the loop to obtain the relationships between factors at adjacent levels and cross levels. The corresponding node pairs in the matrix are determined and connected using lines with arrows from bottom to top. The digraph of the ISM results for the risk factors of fresh agricultural product supply is shown in Fig 1.

The final results of the ISM consist of four levels and 13 key risk factors. The circles with a risk number represent the key risk factors, and the solid lines with an arrow indicate the influence paths. Dashed lines are used to define the range of levels, and L1, L2, L3, and L4 are the hierarchical numbers of the key risk factors identified through ISM.

**MICMAC classification of risk factors.** After the final ISM diagraph is obtained, this paper uses the matrices impact croises multiplication appliqué a un classmen (MICMAC) method to clarify the types of risk factors and ultimately provide preventive measures for different types of risks. According to the driving force and dependence value of each risk factor, each factor is plotted as a point in the four quadrants of the rectangular coordinate system (the X-axis represents the driving force, and the Y-axis represents the dependence), thereby dividing 13 core risk factors into four quadrants, as shown in Fig 2.

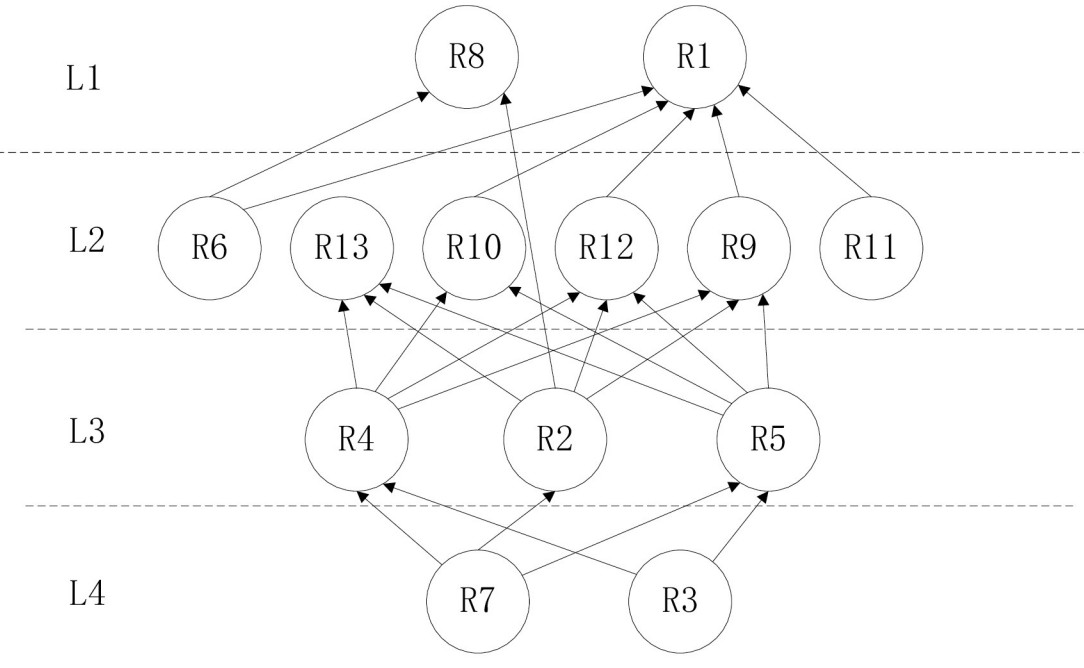

**Fig 1. The ISM results for the risk factors of fresh agricultural product supply.**

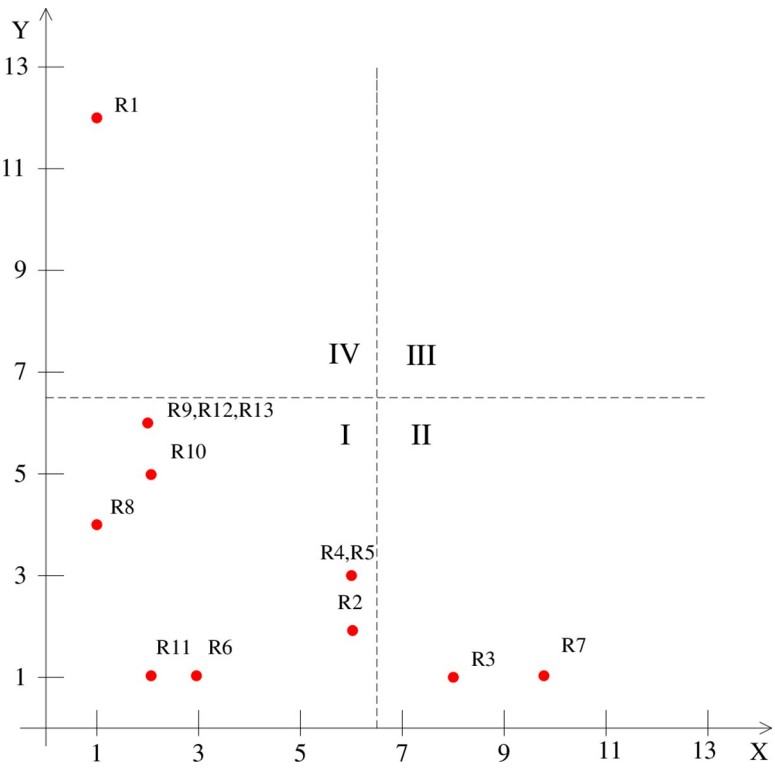

**Fig 2. Classification of the risk factors' dependence-driving force.**

## Analysis of the results

The ISM diagraph and MICMAC classification chart provide insight into the relative importance and interdependencies among various risk factors. Some of the major conclusions of this study are highlighted here.

(1) The set of the top-level risk factors identified by ISM includes completion risk and market risk, indicating that avoiding completion risk and market risk is the ultimate goal of the fresh agricultural product supply. According to the MICMAC analysis, completion risk belongs to the dependent cluster (quadrant IV), and market demand risk R8 belongs to the spontaneous cluster (quadrant I), indicating that the completion risk factor has a strong dependence on other risk factors, while the market risk factor is less dependent on other risk factors. That is, both completion risk and market risk ultimately determine whether the fresh agricultural product supply's ultimate goal can be achieved, but the market risk is more difficult to predict, and completion risk can be traced. Therefore, completion risk should be evaluated in real-time to examine the risk management situation of the fresh agricultural product supply.

(2) The set of second-level risk factors identified through ISM includes regulation risk, pollution risk, biological disease risk, natural climate risk, production risk, and logistics risks. The set of third-level risk factors consists of information distortion risk, technology and management risk, and material equipment risk. The risk factors are all middle-level indirect factors that can indirectly affect fresh produce companies. These risk factors are all middle-level indirect factors that can indirectly affect the fresh agricultural product supply. According to the MICMAC analysis, there are two groups of strongly correlated risk factors in the middle-level risk factors, namely, pollution risk, production risk, and logistics risks at the L2 level, and

technical risk R4 and facility risk R5 at the L3 level. The analysis shows that these internal risk factors interact with each other and amplify the feedback effect. Therefore, treating strongly correlated risk factors as a whole can improve the effectiveness of risk management.

(3) The results of the ISM show that financial risk and artificial risk are at the bottom level, indicating that these two factors are the basic factors affecting the supply of fresh agricultural products. The MICMAC analysis shows that the driving force of artificial risk is the strongest, which indicates that artificial risk is a main source of power in the entire risk transmission system and transmits the influence of low-level factors to high-level factors, as shown through ISM. Financial risk is second only to the artificial risk and plays a role in risk transmission.

(4) According to MICMAC analysis, none of the 13 core risk factors belong to the linked cluster, which indicates that these factors are relatively stable, and changes in any one factor in the system will not affect the other factors. Artificial risk and financial risk belong to independent clusters, indicating that these two factors have a strong influence on other factors but are less affected by other factors. Furthermore, artificial risk and financial risk are not easy to indirectly control by avoiding and mitigating other risk factors, and once these two risks occur, they are likely to cause the formation of a risk chain, which should be noted in the discussion about the key observations of this study. The dependence of regulations risk and natural climate risk is low, indicating that their occurrence is not greatly affected by other risk factors, and they are more difficult to predict and control, but once they occur, the impact on the fresh agricultural product supply is more serious.

## Conclusions

In this study, the social risks and natural risks affecting the fresh agricultural product supply are divided into 13 core risk factors. These risk factors and their hierarchical correlation are discussed by the ISM and MICMAC approaches. The analysis results show that these fresh agricultural product supply risk factors have a strong correlation, and the risk assessment should be carried out from a comprehensive perspective. The suggestions are as follows:

(1) Based on their own situation, fresh agricultural product supply should establish a reasonable risk-sharing mechanism for regulation risk and natural climate risk that are difficult to resist and predict. Through business subcontracting, purchase of insurance and other appropriate control measures, risks are transferred to others or controlled within their own tolerance.

(2) Fresh agricultural product supply should strengthen the improvement of the supervision and error correction system. The financial risk and artificial risk factors that have greater influence and are relatively less affected by other factors are incorporated into the key detection scope, and the role of internal supervision is exerted to correct deviations in time to avoid the formation of risk chains and then affect other risk factors.

(3) Fresh agricultural product supply should avoid the information distortion risk as much as possible. By establishing an information-sharing platform and improving internal information communication channels, fresh agricultural product supply can reduce the dilemma caused by distorted information and untimely information transmissions, such as production decision errors, operational errors, misinterpretation of policies and regulations, untimely storage and transportation, and untimely handling of biological and environmental disasters. In addition, by establishing a good communication mechanism with external demanders, fresh agricultural product supply can avoid or reduce situations in which supply adjustment lags behind market demand, market demand information transmission errors, and information asymmetry between the suppliers and the market.

(4) Based on the characteristics of technology and management risk and material equipment risk that can be easily corrected and the strong correlation between these two risks, fresh agricultural product suppliers should treat these two risks as one and conduct unified management. On the one hand, fresh agricultural product supply needs to develop a comprehensive training system for operating techniques and a set of company management charters. On the other hand, it must ensure the adequacy and quality of transportation vehicles, agricultural equipment, and agricultural materials. Also, technology and management risk and material equipment risk have a strong impact on production risk, logistics risk, pollution risk, and biological disease risk. Therefore, these risks can be reduced indirectly by strengthening the management of technology and management risk and material equipment risk.

## Limitations and further research

Several limitations and opportunities for model improvement exist. All the core risk factors, and their relationships identified in the present study are plausible. However, the ISM model depends heavily on the opinions of experts, and we cannot eliminate the subjectivity of these experts from the results. Although we collected as much of the literature on the risks of the fresh agricultural product supply as possible, we removed unpublished studies from the sample and focused on a higher level of research. We also carefully summarized and reorganized the 13 core risk factors based on experts' opinions of the initial variables, but some uncertainty exists. Encoding the textual data analyzed in this article using a method based on grounded theory can more accurately capture the core risk factors. Second, one disadvantage of ISM is that it can only qualitatively analyze the relationships among the risk factors, and it cannot quantify the transmission and influence of the different risks. Therefore, subsequent research needs to use structural equation models, neural networks and other methods to establish quantitative models; these are fruitful opportunities for future research. Finally, while we put forward some conclusions regarding the general risk management of China's fresh agricultural product supply, specific risk management should still be formulated based on the actual situation.

## Author Contributions

**Methodology:** Xiaoyu Bian, Guohong Shi.

**Resources:** Xiaoyu Bian.

**Software:** Xiaoyu Bian, Guanxin Yao.

**Supervision:** Guanxin Yao.

**Validation:** Xiaoyu Bian.

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
