## [Editor Report · Decision Letter 0]

30 Mar 2020

PONE-D-20-07597

Social and Natural Risk Factors Correlation in China's Fresh Agricultural Products Supply

PLOS ONE

Dear Dr. Bian,

Thank you for submitting your manuscript to PLOS ONE. After careful consideration, we feel that it has merit but does not fully meet PLOS ONE’s publication criteria as it currently stands. Therefore, we invite you to submit a revised version of the manuscript that addresses the points raised during the review process.

We would appreciate receiving your revised manuscript by May 14 2020 11:59PM. To enhance the reproducibility of your results, we recommend that if applicable you deposit your laboratory protocols in protocols.io, where a protocol can be assigned its own identifier (DOI) such that it can be cited independently in the future. For instructions see: http://journals.plos.org/plosone/s/submission-guidelines#loc-laboratory-protocols

We look forward to receiving your revised manuscript.

Kind regards,

Bing Xue, Ph.D.

Academic Editor

PLOS ONE

Additional Editor Comments (if provided):

1) the cover letter should be rewrote. First of all, the journal you submitted is PLOS ONE, not "“System Dynamics Review”. And, please spell the journal name in a right way, it should be "PLOS" instead of "POLS". Hope you never make this mistake again.

2) the order of the second Affiliation should be "School of Business, Yangzhou University" instead of "Yangzhou University School of Business", please check it carefully.

3) a clear statement and describition of the selection process regarding the risk factors is needed in section 4.2/

4) please read the "Guidelines for authors" carefully and make sure all the formats meet the requirement of PLOS ONE. for example, there should be "1, 2, 3……“ after "5. Research methods", you can use either "(1), (2)" or "5.1"

Journal Requirements:

https://doi.org/10.1016/j.frl.2020.101491

In your revision ensure you cite all your sources (including your own works), and quote or rephrase any duplicated text outside the methods section. Further consideration is dependent on these concerns being addressed.
---

## [Author Response · Author response to Decision Letter 0]

2 Apr 2020

Dear Editor:

Thank you for your letter and the comments concerning our manuscript entitled “Social and Natural Risk Factors Correlation in China's Fresh Agricultural Products Supply”. Those comments are all valuable and very helpful for revising and improving our paper, as well as the important guiding significance to our researches. We have studied comments carefully and have made correction which we hope meet with approval. The main corrections in the paper and the responds are as flowing:

In the Editors’ Comments

Some of the major weaknesses of the paper that stand out:

1) the cover letter should be rewrote. First of all, the journal you submitted is PLOS ONE, not "“System Dynamics Review”. And, please spell the journal name in a right way, it should be "PLOS" instead of "POLS". Hope you never make this mistake again.

Response: This is a serious mistake, and we have rewritten the letter, hoping to get your forgiveness.

2) the order of the second Affiliation should be "School of Business, Yangzhou University" instead of "Yangzhou University School of Business", please check it carefully.

Response: We created a Title Page document and wrote it according to the template provided by PLOS ONE. The following is a partial interception of the modified content:

1School of Management, Jiangsu University, Zhenjiang, Jiangsu, China

2School of Business, Yangzhou University, Yangzhou, Jiangsu, China

Other changes can be reviewed in the Title Page document.

3) a clear statement and describition of the selection process regarding the risk factors is needed in section 4.2

Response: We rewrite the selection process for risk factors and make the statement clearer by listing the selection steps.

(1) Collecting relevant data for textual analysis, extracting information according to keyword risk and failure, and determining the initial risk factors.

(2) Identifying members of the expert group. 

(3) Gathering individual opinions on the initial risk factors and analyzing these opinions from the expert group.

(4) Compiling information on new comprehensive opinions and sending to each members of the expert group for review. 

(5) Asking each expert to revise their own opinions based on the responses from the expert group. If the individual's response is significantly different from that of the group norm, the member needs to provide a rationale for his differing viewpoint.

(6) Analyzing the new individual opinions and returning to the members the responses. 

(7) This analysis process is repeated many times to gradually obtain a more consistent result. 

4) please read the "Guidelines for authors" carefully and make sure all the formats meet the requirement of PLOS ONE. for example, there should be "1, 2, 3……“ after "5. Research methods", you can use either "(1), (2)" or "5.1"

Response: we read the "Guidelines for authors" carefully and make sure all the formats meet the requirement of PLOS ONE. For example, we re-edited the format of headings, references, and table headings.

In the Journal Requirements

1. Please ensure that your manuscript meets PLOS ONE's style requirements, including those for file naming. The PLOS ONE style templates can be found at http://www.plosone.org/attachments/PLOSOne_formatting_sample_main_body.pdf and http://www.plosone.org/attachments/PLOSOne_formatting_sample_title_authors_affiliations.pdf.

Response: The two URLs given here cannot be opened. We found the main body and title page format requirements in the PLOS ONE’s submission guidelines, and we read the guidelines carefully and make sure all the formats meet the requirement of PLOS ONE.

Response: We took your suggestion and used AJE's language polishing service.Language changes can be viewed in the new manuscript.

3. We noticed you have some minor occurrence of overlapping text with the following previous publication(s), which needs to be addressed. Your revision ensure you cite all your sources (including your own works), and quote or rephrase any duplicated text outside the methods section. Further consideration is dependent on these concerns being addressed.

Response: we added the previously missed reference and details as follows:

23. Yu, S., Lizhen, C., Huaping, S., Farhad, T. H.. Low-carbon financial risk factor correlation in the belt and road PPP project. Finance Research Letters. 2020 [3 July 2020], Available from: https://doi.org/10.1016/j.frl.2020.101491. 

We tried our best to improve the paper and made changes in the manuscript. We appreciate for your warm work earnestly, and hope that the correction will meet with approval.

Sincerely yours, Xiaoyu Bian

---

## [Decision Letter · Decision Letter 1]

17 Apr 2020

PONE-D-20-07597R1

Social and Natural Risk Factor Correlation in China's Fresh agricultural product supply

PLOS ONE

Dear Dr. Bian,

Thank you for submitting your manuscript to PLOS ONE. After careful consideration, we feel that it has merit but does not fully meet PLOS ONE’s publication criteria as it currently stands. Therefore, we invite you to submit a revised version of the manuscript that addresses the points raised during the review process.

We would appreciate receiving your revised manuscript by Jun 01 2020 11:59PM. To enhance the reproducibility of your results, we recommend that if applicable you deposit your laboratory protocols in protocols.io, where a protocol can be assigned its own identifier (DOI) such that it can be cited independently in the future. For instructions see: http://journals.plos.org/plosone/s/submission-guidelines#loc-laboratory-protocols

We look forward to receiving your revised manuscript.

Kind regards,

Bing Xue, Ph.D.

Academic Editor

PLOS ONE

Reviewers' comments:

Reviewer's Responses to Questions

**Comments to the Author**

1. If the authors have adequately addressed your comments raised in a previous round of review and you feel that this manuscript is now acceptable for publication, you may indicate that here to bypass the “Comments to the Author” section, enter your conflict of interest statement in the “Confidential to Editor” section, and submit your "Accept" recommendation.

Reviewer #1: All comments have been addressed

Reviewer #2: All comments have been addressed

2. Is the manuscript technically sound, and do the data support the conclusions?

Reviewer #1: Yes

Reviewer #2: Yes

3. Has the statistical analysis been performed appropriately and rigorously? 

Reviewer #1: Yes

Reviewer #2: Yes

4. Have the authors made all data underlying the findings in their manuscript fully available?

Reviewer #1: Yes

Reviewer #2: Yes

5. Is the manuscript presented in an intelligible fashion and written in standard English?

Reviewer #1: Yes

Reviewer #2: Yes

6. Review Comments to the Author

Reviewer #1: The manuscript entitled “Social and Natural Risk Factor Correlation in China's Fresh agricultural product supply” has some strengths. This topic is fascinating, considering the increasing importance of social and natural risk factors in the agricultural sector. The paper is well written and easy to follow.

The authors have made a significant effort in revising the paper. They have solved almost all the comments. The paper can be accepted for publication in its current form.

Sincerely

Reviewer #2: (No Response)

7. PLOS authors have the option to publish the peer review history of their article (what does this mean?). If published, this will include your full peer review and any attached files.

Reviewer #1: Yes: Dhekra Ben Amara

Reviewer #2: No

---

## [Author Response · Author response to Decision Letter 1]

20 Apr 2020

Thank you for your recognition of this manuscript entitled “Social and Natural Risk Factor Correlation in China's Fresh agricultural product supply”, we are very pleased that the article has been approved by experts.

---

## [Editor Report · Decision Letter 2]

23 Apr 2020

Social and Natural Risk Factor Correlation in China's Fresh agricultural product supply

PONE-D-20-07597R2

Dear Dr. Bian,

We are pleased to inform you that your manuscript has been judged scientifically suitable for publication and will be formally accepted for publication once it complies with all outstanding technical requirements.

With kind regards,

Bing Xue, Ph.D.

Academic Editor

PLOS ONE
---

## [Editor Report · Acceptance letter]

28 Apr 2020

PONE-D-20-07597R2 

Social and Natural Risk Factor Correlation in China's Fresh agricultural product supply 

Dear Dr. Bian:

I am pleased to inform you that your manuscript has been deemed suitable for publication in PLOS ONE. Congratulations! Your manuscript is now with our production department. 

With kind regards,

on behalf of

Professor Bing Xue 

Academic Editor

PLOS ONE